# Drying Habanero Pepper (*Capsicum chinense*) by Modified Freeze Drying Process

**DOI:** 10.3390/foods9040437

**Published:** 2020-04-05

**Authors:** Cicerón González-Toxqui, Álvaro González-Ángeles, Roberto López-Avitia, David González-Balvaneda

**Affiliations:** 1Facultad de Ingeniería, Universidad Autónoma de Baja California, Blvd. Benito Juárez s/n, Mexicali 21280, B.C., Mexico; ciceron.gonzalez@uabc.edu.mx (C.G.-T.); ravitia@uabc.edu.mx (R.L.-A.); 2Instituto Tecnológico de Mexicali, Ave. Tecnológico s/n Colonia Elías Calles Cp., Mexicali 21376, B.C., Mexico; gonzadav@itmexicali.edu.mx

**Keywords:** dehydration, conserving vegetables, improving shelf-life, rehydrated pepper, histological preparation, green practices

## Abstract

Freeze drying process was applied to habanero pepper and modified, in order to reduce energy expenditure on frozen and dehydration techniques. Six alkaline solutions, olive oil, avocado oil, coconut oil, grape oil, sesame oil and safflower oil, were used to reduce time on vacuum chamber. Also, frozen step was modified by using dry ice (CO_2_) obtaining 43% of energy saving. The final product had high quality, moisture within 3% to 7% range, low microorganisms number, without organoleptic attributes damage and having all the characteristics of a fresh product by rehydrating. Dried sample was rehydrated by immersion in water at 40 °C for 5 min, obtaining 75% of initial humidity.Markedchanges on rehydrated final product was not perceived. The most effective oil to reduce the moisture was safflower followed by coconut and sesame, whilst the least effective were olive, followed by avocado and grape oils.

## 1. Introduction

Mexico stands out in the generation of chili varieties in the world. Around 90% of chili consumed worldwide is of Mexican origin. Other producing countries are China, Indonesia, Turkey, Spain, the United States, and Nigeria. Additionally, 80% of the production of habanero pepper is marketed as fresh and the remaining 20% goes to the preparation of sauces and pasta, oris dehydrated. It exports mainly to the United States, Japan, South Korea, Italy, and Germany [1,2].

Habanero pepper is an herbaceous plant or shrub, branched, reaching a size up to 2.5 meters high. The immature specimens of habanero peppers are green, but their color varies at maturity. The most common colors are orange (semi-ripe) and red when ripe. According to scientific research, the origins of habanero pepper are found in the zone that extends from southern Brazil to northern Argentina, through eastern Bolivia and western Paraguay [3].

Likewise, due to its different properties, habanero pepper is used in different areas as gastronomy, medicine within which its components are used to make ointments that relieve the severe pain caused by arthritis; within the chemical industry, it is used to make the base of some paintings, as well as to make tear gas [4].

Due to the fact that fresh chili is perishable, it is necessary to extend its shelf life by choosing suitable preserving methods, minimizing loss or damage of active ingredient, and consuming the least energy in the process.

There are a lot of drying methods (sun drying, hot air drying, spray drying, vacuum drying, freeze-drying, hybrid methods among others) used to decrease microbial activity, to storage for a long time, and transport the product around the world [5,6,7,8,9].

All methods have advantages and disadvantages, some are cheap andeco friendly and others are expensive like freeze-drying; however, freeze-dryinghas a peculiarity. The final dehydrated product could be rehydrated again, being almost as good as the fresh product.This peculiarity is because the dried product keeps its original shape, something that does not happen in other methods, where the final dried product shrinks.

In literature, there isa lot ofresearch where the freeze-dried method was modified for different aims, for instance, to investigate the effect of various cryoprotectants on cell viability during freeze-drying [10], to assess physico-chemical properties and the antioxidant profile of oyster mushrooms, where the transparent lid of the drying chamber was covered with aluminium foil to prevent the degradation of antioxidants by light oxidation [11]. Another investigation was realized to evaluate the application of CO_2_ laser microperforations to blueberry skin. Under the same set of freeze-drying conditions, blueberries with and without perforations were processed. The results showed that the primary drying time was significantly reduced from 17 ± 0.9 h for nontreated berries to 13 ± 2.0 h when nine microperforations per berry fruit were made. Concomitantly, the quality was also significantly improved, as the percentage of nonbusted blueberries at the end of the process increased from an average of 47% to 86%. It was demonstrated that CO_2_-laser microperforation has high potential as a skin pretreatment for the freeze-drying of blueberries [12].

The aim of this studyis to modify the freeze-drying methodto obtain dried productsin a faster and cheaper way, contributing to green practices.

## 2. Materials and Methods

Fresh habanero pepper was obtained in the local market. The specimens were carefully selected by good condition, no punched or damaged surface, showing good quality evidence. The fresh product was cleaned with water and dried at room temperature.

To determine percent solids and weight (in grams) of the sample as well its moisture percent, a thermo balance was used (Ohaus MB-35 model with halogen source heating). The specimen was heated at 105 °C for 15 min (fresh samples) and 3.5 min (freeze-dried samples) [13]. 

Six alkaline emulsions were prepared (olive oil, avocado oil, coconut oil, grape oil, sesame oil, and safflower oil) by mixing 1.0 liter of alkaline water (pH 10), 50 g of K_2_CO_3_ (with 98% of purity, ASC reagent, Aldrich Co., Toluca, Mexico) and 5 g of olive oil, avocado oil, coconut oil, grape oil, sesame oil, and safflower oil. The final emulsion obtained pH 12. All samples were immersed in pH 12 solution (above described) for 1 minute at room temperature [14,15,16,17].

The freeze-drying process equipment for the test was the Sanshon equipment model FDG-O.5, with 31 × 54 cm process area, 0.5 m^2^ of drying area, and 5 liters of water capacity every 8 h in the drying process. It has a vacuum pump of 6 × 10^−1^ mbar model TRIVAC D16B, 3-ph, 240-265/415-480 V and 60 Hz. It also contains a vacuum chamber, equipped with three heating plates and three shelves where the specimens are placed, a freezing chamber (−40 °C), and a water-heating system on plates (105 °C).

The dehydration process on industrial equipment was performed by 3 stages: first, the product was frozen for 3 h at −35 °C. Once the sample is frozen, a vacuum pressure of 6 × 10^−1^ mbar is applied. Later, two temperature ramps were applied automatically: one for the sublimation of frozen water at 80 °C for 7 h and another for drying the remnant water at 60 °C for 13 h (total drying time 20 h for untreated samples) [18].

In order to reduce energy expenditure on dehydration processes, six pretreatments solutions were used, as well as the frozen stage changed by using dry ice (CO_2_) [6].

The samples were submerged on alkaline solutions for one minute after they passed to the frozen stage for 10 min at −79 °C using dry ice (CO_2_). Once the samplewas frozen, a vacuum pressure of 6−10^−1^ mbar was applied. Then, two temperature ramps were applied automatically, one for sublimation at 80 °C for 7 h and another at 60 °C for 6 h.

The dried sample was rehydrated by immersion in water at 40 °C for 5 min.

The histological cutting preparation was made following the next steps:

Fixation: treat the tissue with chemical substances to keep the properties of the organic tissue intact, inactivating enzymes that degenerate the tissue.Washing: The excess fixative is removed. Clarification: After dehydrating the tissue, a liquid paraffin solution is passed as an inclusion medium and xylene or xylol as a miscible medium [19].Inclusion: The tissue sample is placed in a container and molten paraffin is added at 60 °C, placing the sample in an oven 30 min at 60 °C for 6 h.Cut: The cube obtained can be cut into sections thin enough to allow the passage of light, having a thickness of 5–10 micrometers.Mounting: Using a mounting solution, 1% gelatin at 38 °C, using it as an adhesive for the strip obtained from cutting on a slide.Coloring: To be observed under a microscope, it must be colored or contrasted [20].

Observation of alkaline emulsion penetration on sample.

Alkaline emulsion of ethyl oleate with pH of (12) changes of color when phenolphthalein 1% is added. This approach could help to visually assess the penetration of the solution (alkaline emulsion of ethyl oleate) in grapes. 

Habanero samples were immersed in alkaline emulsion of ethyl oleatefor 1 min. Then, 500 µl of phenolphthalein 1% was added to samples. Images were taken under a dissecting microscope (Stemi DV4, Carl Zeiss, Germany)

For bacterial analysis, serial dilutions of 1:10 of samples in distillated water were performed and 0.1 mL of each dilution was extended in a plate; the plates were incubated until the colonies were appreciable for counting.

The bacteria were isolated in pure culture (standard), in a Petri box and incubated for a period of 24 h at 45 °C.The board was read and the total colony forming units were counted. The specimens were analyzed in an optic microscope (Carl Zeiss, Germany, under visible light).

To observe the morphology of the studied sample, the scanning electronic microscope JEOL JSM- IT 100 (JEOL, Peabody, MA, USA was used. Microchemical analysis of the studied sample was performed by energy-dispersive X-ray spectroscopy (JEOL, Peabody, MA, USA).

## 3. Results and Discussion

After completing the dehydration process for the six different samples, the final moisture percentage was assessed and compared in Table 1.

The initial moisture of the pepper was between 84%–87%. The dehydrated samples obtained moisture 3%–7% range, which is the most cited in the literature [21].It is important to mention that to reach the final time where the sample obtains the desired humidity (3%–7%), that is, the drying process was stopped every hour less than the original time, after which the moisture of the sample was measured, and if the moisture was too great, we kept drying to obtain values inside the range.

Since most of the emulsions presented similar drying times, their effectiveness was sought in the final moisture of the sample, whichis calculated under the next formula: Effectiveness= 1 − (drying process × final moisture/100). It can be seen in Table 1 that themost effective oil to reduce moisture is safflower, followed by coconut and sesame, whilst the least effective are olive, followed by avocado and grape oils.

The energy saving was calculated in relation to the total time of dried sample with treatment. Comparing the total time on the freeze-drying process of sampleswithout pretreatment against olive oil, avocado oil, coconut oil, grape oil, sesame oil, and safflower oil, it observes an energy saving of 36% to 43%. The best alkaline pretreatment for saving energy was coconut oil with a 43%, safflower oil beingable to contribute the least to save energy.

The reduction in drying time protruded with the alkaline solution, probably due to the alkaline solution breaking the wax on the skin surface that acts as a control of moisture diffusion in leaves and fruits, allowing quicker extraction of the water [14].

On the other hand, it is important to mention that the frozen stage must be on a quick mode of freezing; smaller ice crystals will form inside and, therefore, there will be less damage in the wall cells. Due to the above mentioned facts, the freeze-drying process was modified in the first stage, using dry ice (CO_2_) for frozen biological samples [22,23].

Figure 1 shows habanero pepper sections obtained after being immersed in alkaline solution and phenolphthalein in order to determine the penetration of alkaline solution in biological samples. Magenta color reveals the presence of alkaline solution in Figure 1; it can be appreciated that there is no alkaline solution penetration, and the solution remains on skin surface(exocarp), with no impact on habanero pepper endorcarp and mesocarp, just in the cover skin [19].

Due to the hygroscopic property on freeze-dried products, the product was carefully packaged and sealed after the freeze-drying process.

After the freeze-drying process, the organoleptic properties (taste, odor, color, and appearance) show optimal conditions for human consumption. Samples have a typical strong flavor and extremely concentrated odor, more than the fresh samples. It has a slightly foamy appearance, crispy consistency, and intense color, as shown in Figure 2.

Figure 3 shows the energy-dispersive X-ray analysis. It can be seen that the spectra are composed mainly of carbon and oxygen, follow bypotassium and chlorine, and little quantities of phosphorus and sulfur (in this hierarchy, according to peak intense) [24,25].

The taste on final freeze-dried product, using pretreatments solutions, can be evaluated as well; comparing flavors between products is highly important, which is vital for a good-quality product.

Once dehydrated, habanero pepper should be stored properly for quality properties. During storage, a variation of the carotenoid pigments of the skin occurs due to an oxidation process. This is increased by the action of external agents of a physical nature such as temperature, humidity, and light, or chemical nature such as metal ions, enzymes, peroxides and free oxygen. These variations affect essentially the color, odor, composition and visual appearance. On the other hand, these changes may also affect the microbiological properties of the product; the isolation on these factors can be useful to ensure product quality, storage-selecting conditions can predict the useful product life evolution, and avoiding moisture in storage must be considered [18,26].

It is important to remember that any fresh product subjected to freezing and thawing does not have the same fresh look, as there is a broken cell membrane by the crystal-freezing process. Figure 4 shows how the freeze-drying process helps to decrease the microbial load number. Low microbial number of microorganisms is a particular interest for the food industry, due to the fact that lyophilized products could be used in hospitals and proportioned to patients with low defenses in childhood, and seniors [18].

Finally, dried samples were rehydrated by immersion in water at 40 °C for 5 min, obtaining a humidity of 75%,very similar tothe initial humidity of fresh habanero samples(84%–87%). Changes on rehydrated final product were not perceived [27].The idea of rehydration is due to customers consuming sliced pepper in salads or cooking it in pieces and not only in powder form as a dried condiment.

## 4. Conclusions

This investigation shows six pretreatment alkaline solution developments and a CO_2_ frozen method on the habanero pepper freeze-drying process as an alternative, adding improvement generating 36% to 43% in energy savings. The best alkaline emulsion to reducedrying-time was coconut oil. On the other hand, the most effective oil to reduce the moisture was safflower, followed by coconut and sesame, whilst the least effective were olive, followed by avocado and grape oils.

Final moisture for dried samples obtained 3% to 7% after the freeze-drying process. The humidity achieved for rehydrated samples was 75%,similarto the initial humidity of fresh habanero samples. 

After the freeze-drying process, products have hygroscopic properties and high considerations on storage conditions must be applied, such as storage in a cool, dry place away from sunlight, sealed vacuum packing; thus, properties will be preserved to eitherrehydrationorconsumption. Finally, if all enterprises involved in green manufacturing take into account these findings and constantly improve their processes, they will stop emitting several kg of pollutioninto the atmosphere.

## Figures and Tables

**Figure 1 foods-09-00437-f001:**
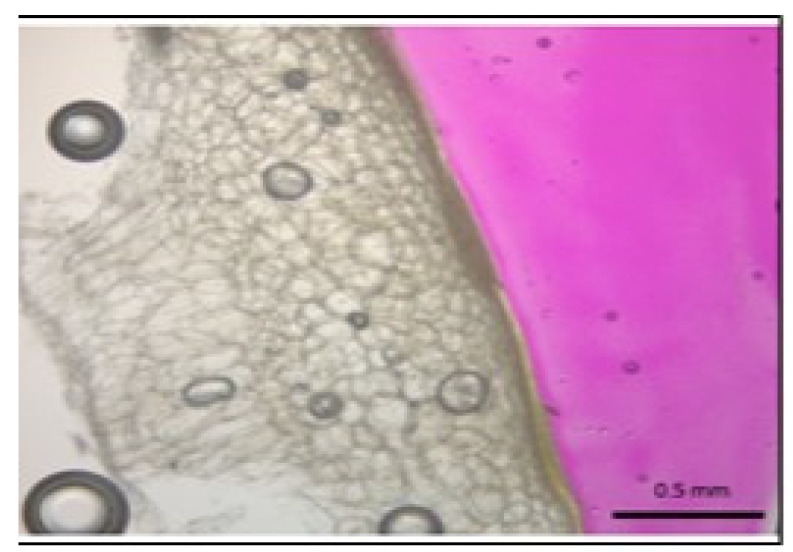
Dried sample section obtained after being immersed in the alkaline solution and phenolphthalein in order to determine the penetration of the alkaline solution. The phenolphthalein turned bright magenta with alkaline solutioncontact; structure viewed at 40× magnification microscopy. The figure shows no alkaline solution penetration in the sample.

**Figure 2 foods-09-00437-f002:**
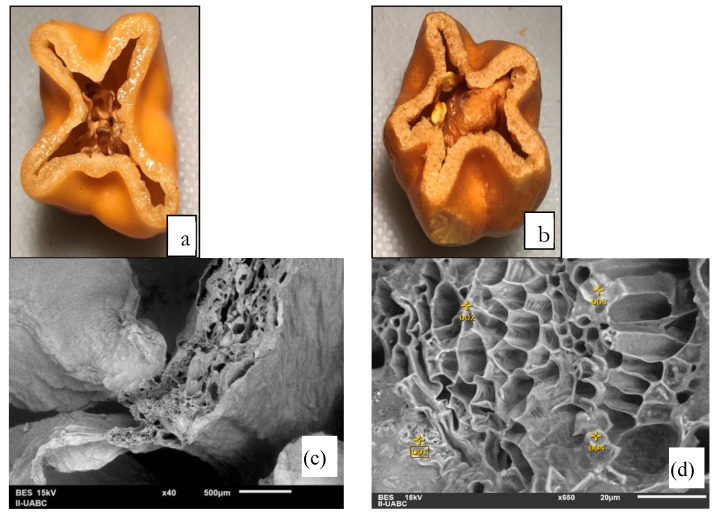
Images of fresh and freeze-dried habanero pepper: (**a**) before freeze-drying, (**b**) after freeze-drying, (**c**) SEM image of dried sample at 40×, (**d**) the place where energy-dispersive X-ray analyses (EDX) were performed (650×) for a dried sample.

**Figure 3 foods-09-00437-f003:**
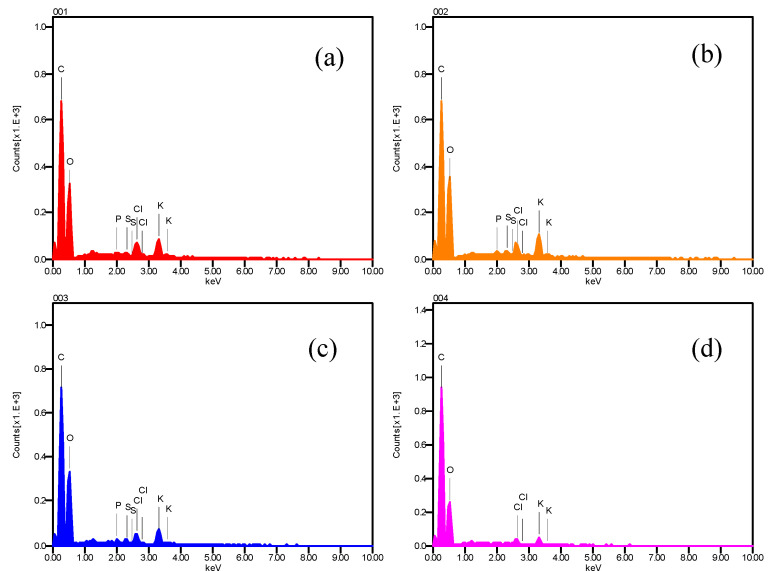
EDX spectrum of atomic elements present in chili, obtained on the dried sample (Figure 2d) with a scanning electronic microscope.(**a**) EDX spectra of atomic elements presents in zone 001 of Figure 2d. (**b**) EDX spectra obtained in zone 002. (**c**) and (**d**) energy-dispersive X-ray spectra obtained in points 003 and 004 of dried sample respectively.

**Figure 4 foods-09-00437-f004:**
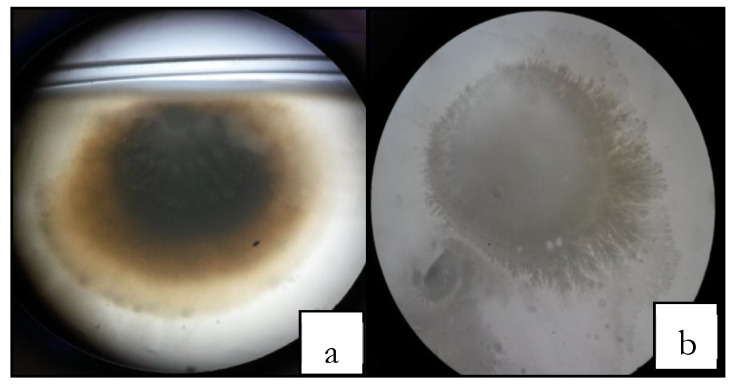
Microbial load habanero pepper sample without alkaline emulsion (**a**), before the freeze-drying process, (**b**)after the freeze-drying process.Structure viewed at 16× magnification microscopy. It can be seen how the microbial load decreases after the sample is dried.

**Table 1 foods-09-00437-t001:** Final freeze time process, drying time process, and final moisture product for treated samples at different alkaline emulsions.

Habanero Pepper Samples	Freezing Process (Minutes)	Drying Process (Hours)	Final Moisture(%)	Energy Saving (%)	Effectiveness of Emulsions (Adimensional)
Samples without pretreatment	180	20	4.6	0	
Olive oil	10	14	6.55	39	0.083
Coconut oil	10	13	4.08	43	0.470
Avocado oil	10	14	6.04	39	0.154
Grape oil	10	14	5.10	39	0.286
Sesame oil	10	14	4.97	39	0.304
Safflower oil	10	14.5	3.01	36	0.564

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
