# Peer review of "Drying Habanero Pepper (Capsicum chinense) by Modified Freeze Drying Process"

_foods, 2020, doi:10.3390/foods9040437_

Round 1

Reviewer 1 Report

The manuscript in the present form is not suitable for publication (major revision). The manuscript needs to be improved, and re-considered for publication. A general comment is that the language needs to be revised. There are a lot of grammatical mistakes throughout the manuscript.

The introduction section should be re-written with the aim of the study to be better explained and justified. Reference on freeze-drying process (or other drying processes) and current approaches are missing. The materials and methods section should be enriched with all the methods used in the manuscript. The table and figure formatting should be checked. The conclusions section should also be improved.

Some remarks:

Introduction: The introduction section focuses on the characteristics of habanero pepper and its health benefits (some parts may be shortened, lines 35-38, 41-55). But there is no information on the processes (e.g. freeze-drying process) and specifically the disadvantages of the commonly used method. The paragraph (Lines 57-60) has been not justified (shelf life extension, minimizing loss or damage of active ingredients, less energy ?).

Materials and Methods:

Lines 62-64: “The specimens were carefully selected 63 by good condition, no cuttings or damage surface, showing a good quality evidence. Washed with 64 water and dried at room temperature after chopped in pieces of 6 mm”. Please correct “Washed…”.

Lines 66-68: Please rephrase(A thermo balance was used for moisture percent evaluation, (Ohaus MB-35 model with halogen 67 source heating), as well as determining percent solids and weight (in grams) of the sample), and add reference (lines 67-68).

Lines 79: Please correct from “Also contains” to “It also contains”. General comment: Check carefully the manuscript for such kind of mistakes.

Lines 100-101: Please write the table caption properly (e.g. not use shows….).

Table 1: Please explain how the energy saving was calculated. Add units for the final moisture (%...).

Lines 106-107: Please rephrase. It is suggested to use “Control samples” instead of “Normal samples”.

Lines 125-127: Please write the figure caption properly, and add the method in the materials and methods section. General comment: Re-write all figure captions (properly) !!! Add all methods presented in the manuscript in the materials and methods section.

Lines 131-133: Please clarify if sensory analysis was conducted.

Figure 4. Please better explain the Figure.

Author Response

Herewith we would like to thank the respected editor and reviewers for providing their precious time and valuable comments in improvisation of our research article

We have addressed all the reviewer comments after careful examination and explained below.

The manuscript in the present form is not suitable for publication (major revision). The manuscript needs to be improved, and re-considered for publication. A general comment is that the language needs to be revised. There are a lot of grammatical mistakes throughout the manuscript.

The introduction section should be re-written with the aim of the study to be better explained and justified. Reference on freeze-drying process (or other drying processes) and current approaches are missing. The materials and methods section should be enriched with all the methods used in the manuscript. The table and figure formatting should be checked. The conclusions section should also be improved.

Some remarks:

Introduction: The introduction section focuses on the characteristics of habanero pepper and its health benefits (some parts may be shortened, lines 35-38, 41-55). But there is no information on the processes (e.g. freeze-drying process) and specifically the disadvantages of the commonly used method. The paragraph (Lines 57-60) has been not justified (shelf life extension, minimizing loss or damage of active ingredients, less energy ?).

Materials and Methods:

Lines 62-64: “The specimens were carefully selected 63 by good condition, no cuttings or damage surface, showing a good quality evidence. Washed with  water and dried at room temperature after chopped in pieces of 6 mm”. Please correct “Washed…”.

Answer: The paragraph was rewrite

Lines 66-68: Please rephrase (A thermo balance was used for moisture percent evaluation, (Ohaus MB-35 model with halogen 67 source heating), as well as determining percent solids and weight (in grams) of the sample), and add reference (lines 67-68).

Answer: The paragraph was rephrased.

Lines 79: Please correct from “Also contains” to “It also contains”. General comment: Check carefully the manuscript for such kind of mistakes.

Lines 100-101: Please write the table caption properly (e.g. not use shows….).

Answer: both mistakes lines 79 and 100 have been corrected

Table 1: Please explain how the energy saving was calculated. Add units for the final moisture (%...).

Answer:  It was divided the total time required to dried the samples with alkaline emulsion by time of samples without alkaline pretreatment

Lines 106-107: Please rephrase. It is suggested to use “Control samples” instead of “Normal samples”.

Answer: the mistake was corrected in manuscript

Answer: It was modified in table 1

Lines 125-127: Please write the figure caption properly, and add the method in the materials and methods section. General comment: Re-write all figure captions (properly) !!! Add all methods presented in the manuscript in the materials and methods section.

Answer: Thanks a lot for the observation. Modifications are reflected in manuscript

Lines 131-133: Please clarify if sensory analysis was conducted.

Answer: It was not sensory analysis conducted

Reviewer 2 Report

The paper presents the effect of habanero pepper pre-treatment on the reduction of freeze-drying energy costs.  Six alkaline emulsions were made to immerse the sample before freeze drying  and the standard frozen method was modified with dry ice.

Comments and suggestions for Authors:

Line 44-45: ‘Due to its different properties, habanero pepper is used in different areas such as gastronomy, medicine and chemical industry [4].’ I suggest Authors to delete this sentence because it is a repetition of  the text in Line 35-38.

Line 47-55: What does this have to do with the research topic presented? Lack of text consistency. Capsaicin hasn’t been studied!

Section 2. ,Materials and Methods’ is unclear ,needs improvements. I suggest that Authors should first presented order of opperations related to habanero pepper processing. And then describe the steps and  the analyzes performed.

In addition, section 2 does not present all research methods (SEM, EDX, microbiology etc.), although results are presented in section 3.

Line 67-68: ,The specimen was heated at 105 ° C for 15 minutes (fresh samples) and 3.5 minutes (freeze dried Samales).’ Why were the samples heated in such a short time? Please enter the literature used to select the heating time? As a rule, a time of at least 2 hours is used! Final moisture of freeze dried products usually is lower than 3%.

Line 70-73: How were the emulsions created?  Please enter some parameters.

Line 94: ‘Dried sample was rehydrated by immersion in water at 40° C for 5 minutes.’ And? Add what was measured, final result?

Line 100-101: ‘Table 1. Shows final freeze time process, drying time process and final moisture product for treated samples at different alkaline solutions.’ The title of table 1 needs improvement, I suggests title: ‘Final freeze time process, drying time process and final moisture product for treated samples at different alkaline emulsions’.

Titles of Figures 1-4 needs improvements!  We don’t describe changes in the drawings in the title! Titles of Figures 1-4 should be re-writing.

Table 1: Energy saving, how it was measured? Please described in the section 2.

Figure 1: I suggest remove the letter ‘a’ from the image for which specific sample this photo applies?  It would be worth presenting photos for all analyzed samples.

Line 131-132: ,…. optimal conditions for human consumption.’ What about fresh habanero pepper? Organoleptic properties? Have they been studied?

Figure 2a-d: It would be better to present photos of chopped pieces of habanero pepper, according to section 2 Line 64. The habanero pepper was chopped? Or not?

Figure 2e: SEM presented a sample? Which one? It must be written.

Figure 2f:  It should be better to present photo in Figure 3.

Figure 3 and Figure 4: Which sample was presented? It must be written.

Line 174-176: ‘Finally, dried samples were rehydrated by immersion in water at 40° C for 5 minutes, obtaining  a humidity of 75 % as compared with initial humidity of fresh habanero sample. Changes on rehydrated final product it were not perceived [18].’ Not presented results! On what basis it was so stated if no results were presented?

Line 190-192: ,The humidity achieved for rehydrated samples was 75 % as compared with initial humidity of fresh habanero sample.’ Unfounded conclusion, no results presented!

Line 200: , Changes on rehydrated final product it is not perceived.’ Unfounded conclusion, no results presented!

Line 201: ,… A.G.A. performed the statistical analysis…’ Statistical analysis? Not presented !

The article should be better edited. Numerous editorial errors, e.g.:

Line 2: , …HabaneroPepper….’ Two words not one.

Line 15: ,…. using dry ice (CO²) obtaining…’ it should be CO2 not CO²

References should be presented in the text, e.g., [1] or [2,3], or [4–6]. Line 55: ,…. the production of fat. [7][8].’ it should be ,… the production of fat  [7,8].’ and Line 74: ,…[9][10][11].’ It should be…[9-11]. Please edit in the whole article.

Temperature value should be presented as a example: 80 °C, 60 °C not as presented in Line 94: ‘…sublimation at 80 ° C for 7 hours and another at 60° C for 6 hour…’. Please edit in the whole article.

Author Response

Herewith we would like to thank the respected editor and reviewers for providing their precious time and valuable comments in improvisation of our research article

We have addressed all the reviewer comments after careful examination and explained below.

The paper presents the effect of habanero pepper pre-treatment on the reduction of freeze-drying energy costs.  Six alkaline emulsions were made to immerse the sample before freeze drying  and the standard frozen method was modified with dry ice.

Comments and suggestions for Authors:

Line 44-45: ‘Due to its different properties, habanero pepper is used in different areas such as gastronomy, medicine and chemical industry [4].’ I suggest Authors to delete this sentence because it is a repetition of  the text in Line 35-38.

Answer: thanks for the observation. The paragraph was deleted

Line 47-55: What does this have to do with the research topic presented? Lack of text consistency. Capsaicin hasn’t been studied!

Answer: the introduction was modified

Section 2. ,Materials and Methods’ is unclear ,needs improvements. I suggest that Authors should first presented order of opperations related to habanero pepper processing. And then describe the steps and  the analyzes performed.

In addition, section 2 does not present all research methods (SEM, EDX, microbiology etc.), although results are presented in section 3.

Answer: The section 2 was modified and was included all missing methods used

Line 67-68: ,The specimen was heated at 105 ° C for 15 minutes (fresh samples) and 3.5 minutes (freeze dried Samples).’ Why were the samples heated in such a short time? Please enter the literature used to select the heating time? As a rule, a time of at least 2 hours is used! Final moisture of freeze dried products usually is lower than 3%.

Answer: The equipment makes it automatically. The difference between times of heating is because the fresh sample has a lot of water (moisture) and the equipment spend a lot of time drying and burning the sample. On the other hand, the freeze dried samples has 6% of moisture therefore spend less time in burning the sample.

Line 70-73: How were the emulsions created?  Please enter some parameters.

Answer: The procedure is in material and method section

Line 94: ‘Dried sample was rehydrated by immersion in water at 40° C for 5 minutes.’ And? Add what was measured, final result?

Answer: It was measure the final moisture that can reach a dried sample in other word: the freeze dried pecuriarity is that  the final dehydrated product could be rehydrated again (obtaining 75% of initial moisture)  being almost as fresh product.

Line 100-101: ‘Table 1. Shows final freeze time process, drying time process and final moisture product for treated samples at different alkaline solutions.’ The title of table 1 needs improvement, I suggests title: ‘Final freeze time process, drying time process and final moisture product for treated samples at different alkaline emulsions’.

Answer: Thank you, the suggestion was made in manuscript

Titles of Figures 1-4 needs improvements!  We don’t describe changes in the drawings in the title! Titles of Figures 1-4 should be re-writing.

Answer: Modifications were made in titles of figures

Table 1: Energy saving, how it was measured? Please described in the section 2.

Answer:  It was divided the total time required to dried the samples with alkaline emulsion by time of samples without alkaline pretreatment

Figure 1: I suggest remove the letter ‘a’ from the image for which specific sample this photo applies?  It would be worth presenting photos for all analyzed samples.

Answer: Thanks the modification has been done in the picture

Line 131-132: ,…. optimal conditions for human consumption.’ What about fresh habanero pepper? Organoleptic properties? Have they been studied?

Figure 2a-d: It would be better to present photos of chopped pieces of habanero pepper, according to section 2 Line 64. The habanero pepper was chopped? Or not?

Answer: Modifications in manuscript were made. Peper were not choped

Figure 2e: SEM presented a sample? Which one? It must be written.

Answer: Thanks,  SEM image of dried sample  at 40 x

Figure 2f:  It should be better to present photo in Figure 3.

Answer: some pictures were erased for better understanding

Figure 3 and Figure 4: Which sample was presented? It must be written.

Answer: Thank you. The fugures titles were modified

Line 174-176: ‘Finally, dried samples were rehydrated by immersion in water at 40° C for 5 minutes, obtaining  a humidity of 75 % as compared with initial humidity of fresh habanero sample. Changes on rehydrated final product it were not perceived [18].’ Not presented results! On what basis it was so stated if no results were presented?

Line 190-192: ,The humidity achieved for rehydrated samples was 75 % as compared with initial humidity of fresh habanero sample.’ Unfounded conclusion, no results presented!

Answer:The paragraphs was rephrased and added the missing data

Line 200: , Changes on rehydrated final product it is not perceived.’ Unfounded conclusion, no results presented!

line 201: ,… A.G.A. performed the statistical analysis…’ Statistical analysis? Not presented !

Answer; sentence was erased

The article should be better edited. Numerous editorial errors, e.g.:

Line 2: , …HabaneroPepper….’ Two words not one.

Answer: the mistake was corrected

Line 15: ,…. using dry ice (CO²) obtaining…’ it should be CO2 not CO²

Answer: the mistake was corrected

References should be presented in the text, e.g., [1] or [2,3], or [4–6]. Line 55: ,…. the production of fat. [7][8].’ it should be ,… the production of fat  [7,8].’ and Line 74: ,…[9][10][11].’ It should be…[9-11]. Please edit in the whole article.

Answer: the inconsistence was fixed

Temperature value should be presented as a example: 80 °C, 60 °C not as presented in Line 94: ‘…sublimation at 80 ° C for 7 hours and another at 60° C for 6 hour…’. Please edit in the whole article.

Answer:The paragraphs was edited

Reviewer 3 Report

The manuscript is not prepared properly; it requires a lot of attention to be considered for the publication of this journal.

The title could be improved to be more specific, but not necessarily, to think about.

Abstract contains information about results that were not given in the manuscript.

Other keywords are needed.

The introduction is only about demonstrating the advantages of the raw material, the lack of information confirming the use of alkaline solutions and freeze-drying, attempts to improve this process. There is no information on the assessment and comparison of energy expenditure during drying in the available literature.

The sentences in lines (Lines 35-36 and 44-45) are almost the same.

The way of citing literature sources is not compliant with the standard, especially on line 52, 55..

It is strange to use the citation number after citing or describing the figures, e.g. Line 134, 150.

The purpose of six alkaline solutions is not clearly defined; as lipid films?

The methodology only applies to sample preparation and drying method. A description of the experiment plan is needed. No information about the purpose of emulsions production. "All samples were immersed in pH 12 solution (above described) for 1 minute at room temperature" and what was done next? Were the samples drained, dripping from excess emulsion?

Line 83, 92: use units from the SI system.

Lines 87-88: "In order to reduce energy expenditure on dehydration processes, six pretreatments solutions were used as well as frozen stage changed by using dry ice (CO2)" - no data available in the discussion of results.

Lines 85, 93: What was the purpose of using such high temperatures of 80 and 60°C? Why was it higher in the first phase of drying, then lower?

Line 94: "Dried sample was rehydrated by immersion in water at 40 ° C for 5 minutes." Why wasn't room temperature used, for example?

The authors use "alkaline emulsions" and more often "alkaline emulsions" - this has to be standardized throughout the whole manuscript.

Results

Table 1:

Correct the unit "m". Enter the units for the size description (column names) and do not repeat with each result. How was the energy saving calculated? It was worth drying "Natural" (control sample without alkaline substance and CO2) for 13 hours, for comparison.

No standard deviations. What was the repeatability of the results?

Lines 116-117: This information (Lines 116-117) should be in the description of fig 1:

"habanero pepper sections obtained after being immersed in alkaline solution and 116 phenolphthalein, in order to determine penetration of alkaline solution"

If the alkaline solution remains in the skin surface, it means that it remains in the dried product!

This habanero pepper skin is thin, but Authors cannot draw such conclusions from this photo. With what microscope was this picture taken? What is in the picture?

The methodology lacks information on the method of taking pictures of the structure of dried samples using the phenolphthalein..

Fig. 2-3: The methodology lacks information on the EDX spectrum of atomic elements obtained with a scanning electronic microscope.

Fig. 4: These pictures and the description is not clear, what is in these pictures?

Lines 174-176: No presentation of these results.

There are often no spaces.

The conclusions relate to results that are missing from the manuscript ..?

The authors have used a literature (21 sources), which should be refined.

Lines 201-203: Lack of "statistical analysis" ... "All authors have critically revised and contributed to the final version of this manuscript." - ?

Author Response

Herewith we would like to thank the respected editor and reviewers for providing their precious time and valuable comments in improvisation of our research article

We have addressed all the reviewer comments after careful examination and explained below.

The manuscript is not prepared properly; it requires a lot of attention to be considered for the publication of this journal.

The title could be improved to be more specific, but not necessarily, to think about.

Answer: Title was rephrased

Abstract contains information about results that were not given in the manuscript.

Answer: modifications were made in manuscript

Other keywords are needed.

Answer: Keyword was added

The introduction is only about demonstrating the advantages of the raw material, the lack of information confirming the use of alkaline solutions and freeze-drying, attempts to improve this process. There is no information on the assessment and comparison of energy expenditure during drying in the available literature.

Answer:The introduction was modified

The sentences in lines (Lines 35-36 and 44-45) are almost the same.

 Answer: Thanks, modifications were made in manuscript

The way of citing literature sources is not compliant with the standard, especially on line 52, 55..

Answer: References were ordered

It is strange to use the citation number after citing or describing the figures, e.g. Line 134, 150.

The purpose of six alkaline solutions is not clearly defined; as lipid films?

Answer: The reduction in drying time was protruding with the alkaline solution, probably due to the alkaline solution broke the wax on the skin surface that is acting as a control of moisture diffusion in leaves and fruits, allowing quicker extraction of the water. Parragraph added to text

The methodology only applies to sample preparation and drying method. A description of the experiment plan is needed. No information about the purpose of emulsions production. "All samples were immersed in pH 12 solution (above described) for 1 minute at room temperature" and what was done next? Were the samples drained, dripping from excess emulsion?

Answer: after immersed in alkaline solution was freezed and dried

Line 83, 92: use units from the SI system.

Lines 87-88: "In order to reduce energy expenditure on dehydration processes, six pretreatments solutions were used as well as frozen stage changed by using dry ice (CO2)" - no data available in the discussion of results.

Answer: Using CO2 reduce the tame of freezing as can be seen in table 1

Lines 85, 93: What was the purpose of using such high temperatures of 80 and 60°C? Why was it higher in the first phase of drying, then lower?

Answer: after freeze the sample is applied vacuum and heat at 60 °C for sublimation of water frozen and after 80 °C for final moisture

Line 94: "Dried sample was rehydrated by immersion in water at 40 ° C for 5 minutes." Why wasn't room temperature used, for example?

Answer: The rehydration is beter a this temperature

The authors use "alkaline emulsions" and more often "alkaline emulsions" - this has to be standardized throughout the whole manuscript.

Results

Table 1:

Correct the unit "m". Enter the units for the size description (column names) and do not repeat with each result. How was the energy saving calculated? It was worth drying "Natural" (control sample without alkaline substance and CO2) for 13 hours, for comparison.

Answer: Thanks the correction was made in table

No standard deviations. What was the repeatability of the results?

Answer: it was not made a statistical study

Lines 116-117: This information (Lines 116-117) should be in the description of fig 1:

Answer title of figure was modified

"habanero pepper sections obtained after being immersed in alkaline solution and 116 phenolphthalein, in order to determine penetration of alkaline solution"

If the alkaline solution remains in the skin surface, it means that it remains in the dried product!

Answer: The analysis of alkaline penetration was made for samples immersed in alkaline emulsion before drying

This habanero pepper skin is thin, but Authors cannot draw such conclusions from this photo. With what microscope was this picture taken? What is in the picture?

Answer: Images were taken under a dissecting microscope (Stemi DV4, Carl Zeiss, Germany) (information added to ms)

The methodology lacks information on the method of taking pictures of the structure of dried samples using the phenolphthalein..

 Answer: the manuscript was modified in method section.

Fig. 2-3: The methodology lacks information on the EDX spectrum of atomic elements obtained with a scanning electronic microscope.

Answer: Methodology was modified

Fig. 4: These pictures and the description is not clear, what is in these pictures?

Answer: The pictures were modified

Lines 174-176: No presentation of these results.

Answer: the manuscript was  complete restructured

There are often no spaces.

The conclusions relate to results that are missing from the manuscript ..?

Answer: the manuscript was  entire restructured

The authors have used a literature (21 sources), which should be refined.

Answer: The references were edited

Lines 201-203: Lack of "statistical analysis" ... "All authors have critically revised and contributed to the final version of this manuscript." - ?

Answer: the sentence was erased

Round 2

Reviewer 1 Report

The modified manuscript has taken into account the suggestions made (technical). However, there are still grammatical mistakes throughout the manuscript ( e.g. lines 147-148: “To observe the morphology of the studied sample was used scanning electronic microscope”). The English language needs to be revised and improved.

The manuscript must be revised.

Some comments:

Please re-write the aim of the study (Lines 61-62: In an attempt to contribute to green practices, it was modified drying method in order to obtain dried product in a faster and cheaper way as well as Lines 55-60: …have a peculiarity….).

Please check Table 1. The captions must be short but informative. Provide the experimental conditions, as far as they are necessary for understanding. A line between samples without pretreatment and olive oil must be deleted.

Add some statistical analysis results (e.g. in Table 1 statistically significant differences between different pretreatments).

Please re-write the Figure 2 caption (avoid repetition).

Conclusions: Please explain: The best alkaline emulsion to reduce time drying was coconut oil and safflower oil obtained the minimum range of acceptance to save time, however it could be used in special cases. Which are the special cases?

Delete “Please add:”  Funding: Please add: “This research was funded by CONAHCYT, grant number 692364/ PhD. scholarship”

Author Response

Some comments:

Please re-write the aim of the study (Lines 61-62: In an attempt to contribute to green practices, it was modified drying method in order to obtain dried product in a faster and cheaper way as well as Lines 55-60: …have a peculiarity….).

Answer: I rewrote the paragraph of the aim

Please check Table 1. The captions must be short but informative. Provide the experimental conditions, as far as they are necessary for understanding. A line between samples without pretreatment and olive oil must be deleted.

Answer: the line was erased

Add some statistical analysis results (e.g. in Table 1 statistically significant differences between different pretreatments).

Answer: A column was added to table 1

Please re-write the Figure 2 caption (avoid repetition).

Answer: the figure 2 caption was rewritten

Conclusions: Please explain: The best alkaline emulsion to reduce time drying was coconut oil and safflower oil obtained the minimum range of acceptance to save time, however it could be used in special cases. Which are the special cases?

Answer: The paragraph was rewritten

Delete “Please add:”  Funding: Please add: “This research was funded by CONAHCYT, grant number 692364/ PhD. scholarship”

Answer: Modifications have been done in manuscript

Reviewer 2 Report

The manuscript is not prepared properly. It requires a lot of attention to be considered for the publication. Example: line 2: Driying HabaneroPepper, line 134: Abanero, line 144:  45 °., line 152: paper, ..., etc.

Units in the manuscript (in the all text) must be improved, example:  60 °C not 60°C, 60% not 60 %.

Line 43-45: In this study, capsaicin-chitosan microspheres were prepared by ionic cross-linking and spray drying by examining the anti-obesity capacity in obese rats [5, 6]. Unclear. What does this have to do with the topic?  It should be written clearly.

Line 51: Osmotic dehydration is type of pretreatment not drying method.

Line 88: Third stage was drying at 60 oC for 6 or 13 h? It should be written clearly.

Line 93-94: Did Authors provide tests without immersing the habanero pepper in emulsions? Only freezing a pepper using dry ice, and them drying? Because it isn’t clearly (energy saving), is it effect of pretreatment (immersing pepper in emulsion) or freezing by using dry ice? Please give details.

Table 1: Why did Authors dry a control sample 20 h instead of 13 h? How was energy saving calculate? It should be written in the text.

Line 163-166: Literature should be cited.

Line 166: leaves?

Line 195-196: Please correct the title of Figure 2. I propose title: Figure 2 Images of fresh and freeze drying habanero pepper: a) before freeze drying, b) after freeze drying, c) SEM image of dried sample at 40x, d) present the place where EDX analyses were performed (650x) for a dried sample.

Line 220: Write [15,25] not [15], [25]

Author Response

Thanks to editor and reviewers for providing their precious time and valuable comments in improving our research article

We have addressed all the reviewer comments

The manuscript is not prepared properly. It requires a lot of attention to be considered for the publication. Example: line 2: Driying HabaneroPepper, line 134: Abanero, line 144:  45 °., line 152: paper, ..., etc.

Answer: Thanks for observation, the mistakes were corrected

Units in the manuscript (in the all text) must be improved, example:  60 °C not 60°C, 60% not 60 %.}

Answer: Thank you for suggestion, the mistakes were corrected

Line 43-45: In this study, capsaicin-chitosan microspheres were prepared by ionic cross-linking and spray drying by examining the anti-obesity capacity in obese rats [5, 6]. Unclear. What does this have to do with the topic?  It should be written clearly.

Answer: the paragraph was erased

Line 51: Osmotic dehydration is type of pretreatment not drying method.

Answer: the paragraph was erased

Line 88: Third stage was drying at 60 oC for 6 or 13 h? It should be written clearly.

Answer: The total time of dried stage in average is 20 hours (7 + 13 = for samples without pretreatment), please apologies, the mistake was corrected.  For pretreated samples was 13 h (7 + 6)

Line 93-94: Did Authors provide tests without immersing the habanero pepper in emulsions? Only freezing a pepper using dry ice, and them drying? Because it isn’t clearly (energy saving), is it effect of pretreatment (immersing pepper in emulsion) or freezing by using dry ice? Please give details.

Answer:  In table 1 it can see the samples without pretreatment and without dry ice (they were made on industrial equipment). The energy saving, was calculated dividing total time of dried sample with treatment by total time of sample without treatment

Table 1: Why did Authors dry a control sample 20 h instead of 13 h? How was energy saving calculate? It should be written in the text.

Answer: Sentence was added to text.

Line 163-166: Literature should be cited.

Answer: reference is cited in text (11)

Line 166: leaves?

Answer:  Leaf and fruit of a plant are covered by natural wax as moisture regulation

Line 195-196: Please correct the title of Figure 2. I propose title: Figure 2 Images of fresh and freeze drying habanero pepper: a) before freeze drying, b) after freeze drying, c) SEM image of dried sample at 40x, d) present the place where EDX analyses were performed (650x) for a dried sample.

Answer: The figure was modified as suggested

Line 220: Write [15,25] not [15], [25]

Answer: The cite was modified as your suggestion

Reviewer 3 Report

The idea of such research using alkaline emulsions and CO2 in the first stage for frozen samples, and also EDX spectrum of atomic elements present in papper using a scanning electronic microscope is very interest, but the manuscript is still not ordered enough, some results are missing. The Authors have tried to correct many issues, mainly improving the introduction keywords and methodology. However, the methodology should be structured, some things are given in too much detail, and it all lacks some sense. Discussed are results that do not relate to this manuscript, I do not understand why, e.g.: Lines 213-220, conclusions..

The experiment plan should be clearly defined. Lack of drying kinetics, on the basis of which it is possible to assess the time of the process; the time of completion of drying, after which the same humidity is obtained.

Tab. 1 - what about using CO2? How were the energy savings calculated? any equation? The energy demand during drying could be measured. That would be more useful.

There are still missing spaces, also in the manuscript title. The method of citing literature sources is still not uniform throughout the manuscript. In some places, parts of the text are marked in colour that has not changed.

Line 86, 95: use units from the SI system.

Line 224, fig. 4: This is not an accurate wording, many microbes are needed in the human body.

Fig. 4 - improve the description of the figure, think about making the designation and interpretation, because you can think that fresh habanero papper is unfit for consumption, washed incorrectly .. ??

The manuscript requires a lot of attention to be considered for the publication of this journal.

I do not recommend publishing the work in the present form.

Author Response

Thanks to editor and reviewers for providing their precious time and valuable comments in improving our research article

We have addressed all the reviewer comments

The idea of such research using alkaline emulsions and CO2 in the first stage for frozen samples, and also EDX spectrum of atomic elements present in papper using a scanning electronic microscope is very interest, but the manuscript is still not ordered enough, some results are missing. The Authors have tried to correct many issues, mainly improving the introduction keywords and methodology. However, the methodology should be structured, some things are given in too much detail, and it all lacks some sense. Discussed are results that do not relate to this manuscript, I do not understand why, e.g.: Lines 213-220, conclusions..

Answer: Some topic like histological preparation was writing in detail by suggestion of reviewers

The experiment plan should be clearly defined. Lack of drying kinetics, on the basis of which it is possible to assess the time of the process; the time of completion of drying, after which the same humidity is obtained.

Answer: Thanks for the observation. To reach the final time where the sample obtained the desired humidity (3-7%), the heuristic method was used. That is, the drying process was stop in random time, after; moisture of the sample was measured and if the moisture was too great we kept drying up to obtain values inside the range.

Tab. 1 - what about using CO2? How were the energy savings calculated? any equation? The energy demand during drying could be measured. That would be more useful.

 Answer: this measurement was not contemplated.

There are still missing spaces, also in the manuscript title. The method of citing literature sources is still not uniform throughout the manuscript. In some places, parts of the text are marked in colour that has not changed.

Line 86, 95: use units from the SI system.

Answer: It was considered to use the SI system, but due to this system use seconds as measure of time the numbers will be very large to specify hours, so we discard the system

Line 224, fig. 4: This is not an accurate wording, many microbes are needed in the human body.

Fig. 4 - improve the description of the figure, think about making the designation and interpretation, because you can think that fresh habanero pepper is unfit for consumption, washed incorrectly .. ??

Answer: Thanks for the observation. The relevant thing about this study is that it can be verified that lyophilization method decreases the microbial activity in the products. Lyophilized products could be used in hospitals and proportioned to patients with low defenses.

The manuscript requires a lot of attention to be considered for the publication of this journal.

I do not recommend publishing the work in the present form.

Round 3

Reviewer 2 Report

Introduction – line 51-55: this topic has been described too briefly. Please give more information about freeze drying (new literature), I propose to extend this paragraph, present the research of other scientists related to the modification this drying method. Also please present the literature confirming conclusions in line 256-258.

How many dryings were carried out for each variant?

How many replicates (final moisture) were done for each sample?

Please explain, how it is known that most energy saving was not affected by freezing in dry ice than immersed material in emulsions?

Line 57: please remove ‘method’. There is drying methods not lyophilization methods.

The manuscript is still not prepared well. There are still numerous mistakes.

Line 2: Driying??? or Drying!

Line 65: remove comma

Line 160, line 171, line 249: Safflower -  write using a small letter

Line 96: Historogical - write using a small letter

Line 109-110: write using a small letter to: xylene, xylol, xilol

Line 112-114: use a small letter to name chemical elements

Line 116-121: units in the manuscript must be improved 60 °C not 60 ° C

Line 150-161: use the same front size as in the rest of the text

Line 192: correct 40x not 40 x

Line 205: remove ‘I propose title: Figure 2’

Line 264, line 267: remove “ and ”

Author Response

Thank you to the respected editor and reviewers for providing their precious time and valuable comments in improvisation of our research article

Introduction – line 51-55: this topic has been described too briefly. Please give more information about freeze drying (new literature), I propose to extend this paragraph, present the research of other scientists related to the modification this drying method. Also please present the literature confirming conclusions in line 256-258.

A= In literature there are a lot of researches where freeze dried method was modified for different aims for instance to investigate the effect of various cryoprotectants on cell viability during freeze-drying [10], to assess Physico-Chemical Properties and the Antioxidant Profile of Oyster Mushrooms, where the transparent lid of the drying chamber was covered with aluminium foil to prevent the degradation of antioxidants by light oxidation [11]. Another investigation was realized to evaluate the application of CO2 laser microperforations to blueberry skin. Under the same set of freeze-drying conditions, blueberries with and without perforations were processed. The results showed that the primary drying time was significantly reduced from 17 ± 0.9 h for nontreated berries to 13 ± 2.0 h when nine microperforations per berry fruit were made. Concomitantly, the quality was also significantly improved, as the percentage of nonbusted blueberries at the end of the process increased from an average of 47% to 86%. It was demonstrated that CO2-laser microperforation has high potential as a skin pretreatment for the freeze-drying of blueberries [12]

How many dryings were carried out for each variant?A=10

How many replicates (final moisture) were done for each sample?A=10

Please explain, how it is known that most energy saving was not affected by freezing in dry ice than immersed material in emulsions?

A=  Untreated samples and non CO2 process time energy = 20h / 1,250 dollars

Treated samples and CO2 process time energy = 13h / 812.50 dollars

(Support by facilities billing information energy which is confidential).

Line 57: please remove ‘method’. There is drying methods not lyophilization methods.

 A= DONE

The manuscript is still not prepared well. There are still numerous mistakes.

Line 2: Driying??? or Drying!A = mistake was corrected

Line 65: remove comma A=mistake was corrected

Line 160, line 171, line 249: Safflower -  write using a small letter=DONE

Line 96: Historogical - write using a small letter=DONE

Line 109-110: write using a small letter to: xylene, xylol, xilol=DONE

Line 112-114: use a small letter to name chemical elements=DONE

Line 116-121: units in the manuscript must be improved 60 °C not 60 ° C=mistake was corrected

Line 150-161: use the same front size as in the rest of the text=DONE

Line 192: correct 40x not 40 x=DONE

Line 205: remove ‘I propose title: Figure 2’=DONE

Line 264, line 267: remove “ and ”=DONE

Reviewer 3 Report

The manuscript is still not prepared properly; it requires a lot of attention to be considered for the publication of this journal.

It is difficult to be sure of the impact of these 6 substances on the freeze-drying process since their share was small (5%). There is no information in the manuscript about the role of these substances, what is the mechanism of their action during drying ..?

Only 1-minute immersion in a solution of pH 12 was used, in addition CO2 was used. Which factor was more important?

In the title, correct "Driying" to "drying"

Lines 96-125: I do not see the need for such an accurate presentation of the preparation description. Just enter the name of the determinations and briefly describe the principle, without unnecessary details and explanations.

Line 152: What about "the heuristic method"? It is not enough to mention it, because there are various methods, which, however, can not be used here.

Lines 158-159: What does "drying process" in the formula Effectiveness mean? What is the unit?

Lines 167-170: If so calculated "The energy saving", then the energy of samples without treatment is 100%! Therefore, the savings should be calculated as 100% - (minus) the value given in the table. Calculations show that this is how it was counted, but the wrong formula was given. It is better to write "The energy saving was calculated in relation to total time of dried sample with treatment .."

Lines 153-155: “That is, the drying process was stop in random time, after; moisture of the sample was measured and if the moisture was too great we kept drying up to obtain values inside the range.” - this behavior disrupted the entire drying process, so the indicators given can be wrong.

Lines 173-175: It is difficult to indicate the effect of added substances, which 5%.

In addition, it is difficult to indicate the effect of the added substances, whose share was only 5%. Especially that the justification wrote about the action of alkaline solution and not this additive.

Lines 198-200: "Samples have typical strong flavor and extremely concentrated odor more than the fresh samples. It has a slightly foamy appearance, crispy consistency and intense color as shown in Figure 2 [21]." - The way of citing "as shown in Figure 2 [21]" is strange. Is Figure 2 made by the Authors, or does it come from an article [21]?

In addition, the crunchiness of dried material cannot be assessed on the basis of the internal microstructure!

Lnes 231-233: "Low microbial number of microorganisms is a particular interest for food industry, due to Lyophilized products could be used in hospitals and proportioned to patients with low defenses childhood, and third age people [18]." - Not all microbes are bad. This sentence makes no sense.

Line 234: Instead of Fig 4, it is better to give the number of colonies of the dried tested samples! What exactly is in these pictures?

Line 253-255: "Freeze dried product can be consumed or used on several ingestion types as dehydrated (food 253 flavoring and coloring), or in aqueous solutions (medicines, active compounds with antimicrobial 254 effect on bacteria, fungi and viruses)."- this is not a conclusion from the study carried out, delete!!! – “antimicrobial effect on bacteria, fungi and viruses”??!

Author Response

Thanks a lot  to the respected editor and reviewers for providing their precious time and valuable comments in improvisation of our research article

It is difficult to be sure of the impact of these 6 substances on the freeze-drying process since their share was small (5%). There is no information in the manuscript about the role of these substances, what is the mechanism of their action during drying ..?

A= The mechanism of these substances is to open the pore of the biological sample, through humidification, if the pore opens  ice will come out faster.

the difference among alkaline substances is 5% but is 40% compared with control samples (samples without alkaline emulsion)

Only 1-minute immersion in a solution of pH 12 was used, in addition CO2 was used. Which factor was more important?

A= BOTH, time and energy saving.

Applying 1-minute immersion= 6h reduction time drying process.

Applying CO2= 3h reduction time frozen process.

In the title, correct "Driying" to "drying" answer: thanks, the mistake was done

Lines 96-125: I do not see the need for such an accurate presentation of the preparation description. Just enter the name of the determinations and briefly describe the principle, without unnecessary details and explanations.=DONE

Line 152: What about "the heuristic method"? It is not enough to mention it, because there are various methods, which, however, can not be used here.=DONE

Lines 158-159: What does "drying process" in the formula Effectiveness mean? What is the unit?

  1. There is a dimensional

Lines 167-170: If so calculated "The energy saving", then the energy of samples without treatment is 100%! Therefore, the savings should be calculated as 100% - (minus) the value given in the table. Calculations show that this is how it was counted, but the wrong formula was given. It is better to write "The energy saving was calculated in relation to total time of dried sample with treatment .."=DONE

Lines 153-155: “That is, the drying process was stop in random time, after; moisture of the sample was measured and if the moisture was too great we kept drying up to obtain values inside the range.” - this behavior disrupted the entire drying process, so the indicators given can be wrong.

A= Typing error, misunderstood translation, correction was done.

Lines 173-175: It is difficult to indicate the effect of added substances, which 5%.

A= Following an established industrial process, whichin this study was follow, it is very difficult not to see these results.

In addition, it is difficult to indicate the effect of the added substances, whose share was only 5%. Especially that the justification wrote about the action of alkaline solution and not this additive.

 A= Following an established industrial process, whichin this study was follow, it is very difficult not to see these results.

Lines 198-200: "Samples have typical strong flavor and extremely concentrated odor more than the fresh samples. It has a slightly foamy appearance, crispy consistency and intense color as shown in Figure 2 [21]." - The way of citing "as shown in Figure 2 [21]" is strange. Is Figure 2 made by the Authors, or does it come from an article [21]?=DONE

In addition, the crunchiness of dried material cannot be assessed on the basis of the internal microstructure!

A= changed figure 2 and reference as suggested

Lnes 231-233: "Low microbial number of microorganisms is a particular interest for food industry, due to Lyophilized products could be used in hospitals and proportioned to patients with low defenses childhood, and third age people [18]." - Not all microbes are bad. This sentence makes no sense.

A=It is not saying that are all bad, it is only mentioned that it is important the low microbial matter in certaintypes of peoplewho consume foodproducts.

Line 234: Instead of Fig 4, it is better to give the number of colonies of the dried tested samples! What exactly is in these pictures?

A= Discoloration of the microbial stain is appreciated, before and after freeze drying process.

Line 253-255: "Freeze dried product can be consumed or used on several ingestion types as dehydrated (food 253 flavoring and coloring), or in aqueous solutions (medicines, active compounds with antimicrobial 254 effect on bacteria, fungi and viruses)."- this is not a conclusion from the study carried out, delete!!! – “antimicrobial effect on bacteria, fungi and viruses”??!

Answer=the word was delete